# Effect of Cigarette and E-Cigarette Smoke Condensates on *Candida albicans* Biofilm Formation and Gene Expression

**DOI:** 10.3390/ijerph19084626

**Published:** 2022-04-12

**Authors:** Farnoosh Haghighi, Leah Andriasian, Nini Chaichanasakul Tran, Renate Lux

**Affiliations:** 1Section of Biosystems and Function, Division of Oral and Systematic Health Sciences, School of Dentistry, University of California Los Angeles, Los Angeles, CA 90095-1668, USA; fhaghighi@g.ucla.edu; 2School of Dentistry, University of California Los Angeles, Los Angeles, CA 90095-1668, USA; leahandriasian@g.ucla.edu; 3Section of Pediatric Dentistry, University of California Los Angeles, Los Angeles, CA 90095-1668, USA; ninic@dentistry.ucla.edu

**Keywords:** biofilm, *C. albicans*, cigarette, e-cigarette, gene expression, HWP1

## Abstract

Smoking triggers environmental changes in the oral cavity and increases the risk of mucosal infections caused by *Candida albicans* such as oral candidiasis. While cigarette smoke has a significant impact on *C. albicans*, how e-cigarettes affect this oral pathogen is less clear. Here, we investigated the effect of cigarette and e-cigarette smoke condensates (CSC and ECSC) on *C. albicans* growth, biofilm formation, and gene expression. Whereas pure nicotine (N) at the minimum inhibitory concentration (MIC, 4 mg/mL) prevented *C. albicans* growth, enhanced biofilm formation was observed at 0.1 mg/mL. In contrast, at this nicotine sub-MIC (0.1 mg/mL) concentration, CSC and ECSC had no significant effect on *C. albicans* biofilm formation. Additionally, N, CSC, and ECSC increased the expression of HWP1 and SAP2 genes. The ECSC group exhibited elevated expression levels of the EAP1 and ALS3 genes, compared to the nicotine-free ECSC (−) control. Moreover, our in vitro study illustrated that the antifungal drugs, fluconazole and amphotericin B, alleviated the effect of nicotine on *C. albicans* gene expression. Overall, the results of the study indicated nicotine from different sources may affect the pathogenic characteristics of *C. albicans*, including hyphal growth, biofilm formation, and particularly the expression of virulence-related genes.

## 1. Introduction

Smoking is a well-established risk factor for oral diseases such as oral cancer, caries, candidiasis, and periodontal diseases [1,2]. Indeed, cigarette smoke has a strong impact on the oral mycobiome communities and especially *C. albicans*, which is the major cause of oral candidiasis [3,4,5]. Cigarette smoke contains numerous toxic chemicals with nicotine as the main component [6,7]. It has been reported that the nicotine concentration in the saliva of smokers varies between 0.07 to 1.56 mg/mL [8,9]. While this amount seems low, it is sufficient to promote severe oral diseases [10,11,12]. In addition to adversely affecting oral tissues, nicotine has a critical effect on the oral microbiome by altering microbial growth, attachment, and biofilm formation of pathogenic microorganisms including *C. albicans* [9,13]. Notably, nicotine affects the expression of the hyphal wall protein 1 (HWP1) and Agglutinin-like sequence 3 (ALS3) genes which are known virulence factors of *C. albicans* and could be at least partially involved in the observed effects on growth, adherence, and biofilm formation [7,14].

Electronic cigarettes (e-cigarettes) have been marketed as a “safer alternative” to combustible cigarettes and have become popular, especially among young adults [15,16]. Contrary to these claims, however, e-cigarettes were reported to induce inflammation and produce destructive free radicals that affect the innate immune system and increase oral infections [17]. While confirmation of the association between e-cigarette smoke and the oral microbiome is developing, recent evidence suggests its impact is distinct from cigarette smoke [18]. An in vitro study reported e-cigarette smoke had a minor albeit statistically not significant effect on the survival and growth of oral commensal streptococci compared with cigarette smoke [19]. Furthermore, exposure to e-cigarette smoke has led to changes in the transcriptional activity of *Streptococcus pneumonia* without altering virulence properties [20]. On the other hand, it has been reported that e-cigarette vapor promoted the growth and expression of virulence genes in *Streptococcus mutans* and elevated biofilm formation on the surface of teeth [21]. Moreover, e-cigarette smoke altered the interaction of *C. albicans* with gingival epithelial cells and increased the growth of *C. abicans* as well as the expression of the secreted aspartyl proteinase (SAP) gene [5]. 

Although the effect of cigarette smoke on *C. albicans* has been explored [7,8,14], the impact of e-cigarettes is less clear. In addition, despite the heightened risk of oral candidiasis for smokers [3,4], it is currently unclear if nicotine affects the efficacy of antifungal treatments. In this study, we investigated the effect of nicotine in cigarette and e-cigarette smoke condensates (CSC and ECSC) sources on *C. albicans* behavior such as growth, biofilm formation, antifungal treatment, and expression of selected virulence-associated genes.

## 2. Materials and Methods

### 2.1. Preparation of CSC and ECSC

Standard IR3F cigarettes (Kentucky Tobacco Research and Development Center, Orlando, FL, USA) and two different e-cigarette liquids containing 0 and 24 mg nicotine (Vaper Vapes Inc., Sand City, CA, USA) were used to prepare cigarette smoke condensates (CSC) and e-cigarette smoke condensates with nicotine (ECSC) and without nicotine, (ECSC (−)), respectively. Cigarettes/e-cigarettes (Weez Inc., Lombard, IL, USA) were inserted into the end of a plastic tube connected to a vacuum flask containing 20 mL water as described previously [22]. The other outlet of the flask was attached to a standard vacuum line. The cigarette/e-cigarette was lit, and smoke was extracted by using the vacuum to pull down the smoke into the water (Figure 1). This process was repeated for 20 cigarettes and 20 mL of the different e-liquids, which resulted in similar nicotine concentrations for CSC and ECSC (3.04 and 2.99 mg/mL, respectively), while nicotine was not detected in ECSC (−) as expected. The concentration of nicotine in the prepared condensates was measured by High-Performance Liquid Chromatography (HPLC 1100 with UV detector at 254 nm, Agilent Technologies, Santa Clara, CA, USA) (Figure A1). The HPLC system was equipped with an Agilent Eclipse XDB-C18, 3.5 μm, 4.6 × 150 mm column, using H2O with 0.1% trifluoroacetic acid as the mobile phase A, and methanol with 0.1% trifluoroacetic acid as the mobile phase B. The gradient was 5–95% mobile phase B over 4 min then held at 95% mobile phase B for 4 min, then returned to 5% mobile phase B for 1 min. The flow rate was 1 mL/min. UV absorbance was detected at 254 nm using a diode array. Pure nicotine (N) (Thermo Fisher Scientific Inc., Wilmington, DE, USA) served as a standard control for all experiments. 

### 2.2. Candida albicans Strain, Culture Conditions, and Treatments

*Candida albicans* SC5314 was grown aerobically at 37 °C on YPD agar or in liquid culture [23]. For liquid planktonic culture growth, one colony of *C. albicans* was inoculated into 10 mL YPD broth. The YPD broth was supplemented with N/CSC/ECSC or ECSC (−) as needed. The culture was grown to the logarithmic growth phase aerobically at 37 °C in a shaker incubator at 200 rpm for 18 h. *Candida albicans* cells were collected, washed with PBS (phosphate-buffered saline) before further experiments. 

### 2.3. Measuring the Effect of N/CSC/ECSC or ECSC (−) on C. albicans Growth (MIC and MFC)

Antifungal susceptibility assessment of *C. albicans* in the presence and absence of nicotine from different sources were performed using YPD broth microdilution method. Serial dilutions of N/CSC/ECSC or ECSC (−) and fluconazole/amphotericin B were prepared in PBS. *Candida albicans* at a concentration of 1 × 10^3^ cells/mL in YPD broth was grown in the presence of 0–16 mg/mL nicotine from different sources or 0–64 µg/mL fluconazole/amphotericin B in 24-well plates at 37 °C for 18 h. The minimum inhibitory concentration (MIC) was considered as the lowest concentration that inhibits growth of *C. albicans*. MICs were confirmed via spectrophotometric evaluation (optical density 600 nm). A 100 µL aliquot from each well was plated onto YPD agar and incubated at 37 °C for 48 h. Minimum fungicidal concentration (MFC), was defined as the lowest concentration (relative to the nicotine content) of N/CSC/ECSC or ECSC (−) that kills the candida cells and resulted in the growth of fewer than two colonies of *C. albicans*.

### 2.4. Measuring the Effect of N/CSC/ECSC+N or ECSC-N on C. albicans Biofilm Formation

Qualitative and quantitative experiments were performed to define the effect of nicotine from different sources on biofilm formation. A suspension of *C. albicans* 1 × 10^6^ cells/mL in YPD was incubated in the presence of, N/CSC/ECSC or ECSC (−), at a sub-MIC concentration of 0.1 mg/mL for 18 h at 37 °C in 24-well plates. Then, the biofilms were gently washed with PBS to remove planktonic cells. The morphology of biofilm cells was assessed via phase contrast microscopy through a 20× objective (LD Plan-Neofluar 20×/0.4) using an inverted microscope (Zeiss, Axio Vert.A1). Cells were then detached from the bottom of each well prior to being resuspended by pipetting and vortexing in 200 μL  PBS for further analysis. *Candida albicans* biofilm formation was assessed by colony-forming units (CFU) and an MTT-based (3-(4,5-dimethylthiazol-2-yl)-2,5-diphenyl tetrazolium bromide) cell viability assay. Briefly, the MTT stock solution (5 mg/mL) was prepared in PBS and added to the wells. The plate was incubated at 37 °C for 4 h in the dark [24], according to the manufacturer’s instructions (Thermo Fisher Scientific Inc., Carlsbad, CA, USA). One hundred μL  DMSO was added to each well to dissolve formazan, the metabolic product of MTT. After 15 min incubation on an orbital shaker, optical density was measured at 600 nm using a spectrophotometer (Thermo Scientific, Wilmington, DE, USA).

### 2.5. Measuring the Effect of N/CSC/ECSC or ECSC (−) on C. albicans Gene Expression

*Candida albicans* planktonic and biofilm cells were cultured in the presence and absence of 0.1 mg/mL nicotine from different sources for 18 h at 37 °C. The *C. albicans* cells were harvested by centrifugation at 12,000 rpm and washed two times with sterile PBS. The supernatants were discarded, and pellets were suspended in 700 µL QIAzol reagent (Qiagen, Dusseldorf, Germany). Glass beads (Life Technologies, 0.1 mm) were added for mechanical disruption of the cells (5 × 1 min, followed by 1 min incubation on ice) using a bead-beater (Bead MILL 24 Fisher Scientific, USA). Following cell lysis, total RNA was extracted from samples using the MiRNA easy Micro kit (Qiagen, Düsseldorf, Germany) according to the manufacturer’s instructions. Residual DNA was removed with the TURBO DNA-free™ Kit (Invitrogen, Thermo Fisher Scientific Inc., Carlsbad, CA, USA). RNA was further purified (Zymo Research RNA clean and concentrator kit) and the concentration was determined using NanoDrop (2000 Thermo Fisher Scientific Inc., Wilmington, DE, USA). RNA was frozen immediately at −80 °C for further use.

### 2.6. Quantitative Real-Time Polymerase Chain Reaction (qRT-PCR)

RNA from each sample (500 ng) was reverse transcribed into cDNA using the SuperScriptIII first-stand RT-PCR kit (Invitrogen, Thermo Fisher Scientific Inc., Carlsbad, CA, USA). The PCR conditions for generating the cDNA templates were 5 min at 25 °C, 1 h at 42 °C, and 5 min at 85 °C. 

The qPCR reactions were carried out using a Bio-Rad iCycler Thermal Cycler with iQ5 Multicolor Real-Time PCR System Detection (iQ 5 RTPCR QPCR, BioRad). The qPCR mixture was prepared by adding 5 mM of each specific primer (Table A1) mixed with 5 μL SYBR Green Master Mix (BioRad) and 3 μL of Nuclease-free water. Each cDNA dilution sample (1 μL, 1:5) was added to 9 μL qPCR mixture. 

Cycling conditions for HWP1 were 3 min at 95 °C, followed by 30 cycles of 10 s at 95 °C, 30 s at 54 °C, and 40 s at 72 °C. For enhanced adherence to polystyrene 1 (EAP1), ALS3, SAP2, and Actin 1 (ACT1) the cycling conditions were as follows: 5 min at 95 °C, followed by 30 cycles of 15 s at 95 °C, 30 s at 60 °C, and 30 s at 72 °C. ACT1 was used as a reference gene for this study. The specificity of primers was determined by the presence of melting temperature peaks. The results were analyzed using the 2−ΔΔCt relative expression method [25].

### 2.7. Statistical Analysis

Each experiment was performed as at least three separate experiments in triplicate. Data are expressed as mean ± standard error of the mean (SEM). Statistical analysis of differences between the control (non-treated) and the test groups (treated) was performed using the two-way ANOVA (multiple comparisons) function in GraphPad (GraphPad Prism version 8.0.0, GraphPad Software, San Diego, CA, USA), and statistically significant differences were defined as follows: * *p* < 0.05, ** *p* < 0.01, *** *p* < 0.001, and **** *p* < 0.0001.

## 3. Results

### 3.1. Minimum Inhibitory Concentration (MIC) and Minimum Fungicidal Concentration (MFC)

MIC and MFC were determined as the lowest concentration of nicotine from different sources that inhibited growth or achieved killing of planktonic *C. albicans* cells (Table 1). The MIC values of *N*, *CSC*, *and ECSC* against *C. albicans* were 4, 0.25, and 0.5 mg/mL and MFC values were 8, 0.5, and 1 mg/mL, respectively (part A of Table 1), whereas ECSC (−) did not show any inhibitory or killing effect on *C. albicans*. Considering the MIC values against *C. albicans* and the inhibitory effect of nicotine at a concentration of 0.25 mg/mL for CSC, a sub-MIC nicotine concentration of 0.1 mg/mL of CSC, ECSC, and N was selected for all experiments in this study. 

In addition, the effect of the antifungal drugs, fluconazole, and amphotericin B, on *C. albicans* was evaluated. The MIC values of fluconazole and amphotericin B against *C. albicans* were 8 and 0.25 µg/mL, and MFC values were 16 and 1 µg/mL (part B of Table 1), respectively. Sub-MIC concentrations of 4 µg/mL fluconazole and 0.1 µg/mL amphotericin B, which did not affect cell viability and growth of *C. albicans* following exposure to N, CSC, ECSC, and ECSC (−) (Figure A2 and Figure A3) were chosen for all subsequent experiments including antifungal drugs.

### 3.2. Candida albicans Biofilm Formation

The effect of N, CSC, ECSC, and ECSC (−) on *C. albicans* biofilm formation after 18 h of incubation in YPD medium was evaluated using colony-forming unit (CFU) counts (Figure 2A) and the MTT assay (Figure 2B). Pure nicotine at the concentration of 0.1 mg/mL significantly enhanced *C. albicans* biofilm formation. Interestingly, CSC, ECSC, and ECSC (−) at the same concentration of nicotine had no statistical difference on biofilm formation (Figure 2A,B). Moreover, *C. albcians* exposed to nicotine sources, N, CSC, and ECSC showed more hyphae form, compared to ECSC (−) and untreated *C. albicans* (controls) which exhibited yeast form (Figure 3). 

### 3.3. The Effect of N, CSC, ECSC, and ECSC (−) on Virulence Gene Expression in C. albicans Biofilms

Since the above results showed that N increased *C. albicans* biofilm formation, we determined whether this observation involve the regulation of certain *C. albicans* genes that were previously described for their role in virulence [22,26,27]. Quantitative RT-PCR demonstrated that HWP1 gene expression was significantly increased (*p* < 0.0001), following exposure of *C. albicans* planktonic cells to N (4-fold), CSC (2-fold), and ECSC (2.5-fold) (Figure 4*)*. In addition, significant differences in the expression of HWP1, SAP2, and EAP1 were identified between the ECSC and ECSC (−) groups (Figure 4).

Overexpression of HWP1 and SAP2 was detected in *C. albicans* biofilms treated with N (3-fold and 2.5-fold, respectively), CSC (2-fold for both), and ECSC (2-fold for both) (Figure 5) compared to the untreated control (−). However, no significant difference in the expression of EAP1 and AlS3 was observed in *C. albicans* planktonic and biofilm cells between the different groups (Figure 5).

### 3.4. The Effect of N, CSC, ECSC, and ECSC (−) on the Expression of Virulence Genes in C. albicans Biofilms Treated with Fluconazole and Amphotericin B

The relative expression of HWP1, EAP1, SAP2, and ALS3 after exposure to 0.1 mg/mL nicotine from different sources as well as in combination with the antifungal drugs, fluconazole and amphotericin B, was determined in *C. albicans* planktonic and biofilm cells (Figure 6 and Figure 7). A significant difference (*p* < 0.0001), between ECSC and the ECSC (−) control in the expression of EAP1 and ALS3 in fluconazole treated *C. albicans* planktonic was observed (Figure 6A). However, *C. albicans* planktonic treated with antifungal drugs after exposure to N/CSC/ECSC or ECSC (−) did not show significant differences in comparison to the control group (Figure 6A,B). 

The expression of ALS3, SAP2, and EAP1 was not significantly increased in *C. albicans* biofilms pretreated with a sub-MIC concentration of fluconazole (4 µg/mL) and exposed to N/CSC/ECSC or ECSC (−) (Figure 7A). Similarly, *C. albicans* biofilm pretreated with a sub-MIC concentration of amphotericin B (0.1 µg/mL) did not demonstrate a significant difference compared to the control group (Figure 7B).

## 4. Discussion

Cigarette smoke has been established as a major environmental factor that affects oral microorganisms such as *C. albicans*, the leading etiological agent of oral candidiasis [28,29,30]. The emergence of novel smoking products, in particular e-cigarettes that have become increasingly popular over the past few years, has led to rising concerns about their effect on oral microbiome health [18,31]. As the effect of e-cigarettes on *C. albicans* has not been fully elucidated, in this study, we investigated and compared for the first time the impact of N, CSC, ECSC, and ECSC (−) on *C. albicans* growth, biofilm formation, and the expression of certain virulence genes.

In this study, the inhibitory (MIC) and killing (MFC) effects of nicotine from different sources were determined and found to be within the range of nicotine levels present in the saliva of smokers [8,9]. Concentrations of N higher than 4 mg/mL showed an inhibitory effect on *C. albicans* (Table 1), which was in agreement with previous reports [8,32]. Interestingly, CSC and ECSC produced inhibitory effects against *C. albicans* at nicotine levels that were about an order of magnitude lower than for N (Table 1). This difference could be related to the other complex component present in the CSC extract such as tar which has been reported to inhibit the growth of oral microorganisms [33,34]. However, these compounds alone do not seem to be very harmful to *C. albicans* as ECSC (−) did not impact its growth. It seems this fungus is more sensitive to nicotine than other components of the extraction, even though this hypothesis needs further investigation.

Colony-forming unit and MTT assay viability measurements revealed that although N at certain concentrations inhibits *C. albicans* growth, lower concentration (0.1 mg/mL) had the opposite effect and enhanced *C. albicans* biofilm formation (Figure 2A,B). In addition, nicotine from different sources enhanced hyphae formation in *C. albicans* biofilms (Figure 3). A similar study supported our data that N increased the biofilm formation of *C. albicans* [32]. In addition, a dose–response effect of CSC on promoting adhesion and biofilm formation of *C. albicans* as well as the inhibitory impact of CSC at higher concentrations was previously reported [3,22]. This seemingly contradictory behavior of *C. albicans* after exposure to CSC has been explained by the mechanism of kinase pathways, which regulate the expression level of key virulence factors including hyphal formation and host interaction. These pathways are elevated by CSC at specific concentrations, though inhibited when the concentration of these compounds is high which decreases *C. albicans* adhesion as well as biofilm formation [3,22]. Moreover, a similar study on oral streptococci found that CSC significantly impacts biofilm growth, while ECSC only resulted in a small non-significant reduction compared to ECSC (−) (compared to the untreated group) on oral streptococci. This raised the argument of the toxic impact of e-cigarettes on the growth of microorganisms [19].

*Candida albicans* morphological changes, growth, attachment, and biofilm formation are controlled by several genes such as HWP1, EAP1, SAP2, and ALS3 [35]. The HWP1 gene encodes the hyphal wall protein1, which is the major protein of *C. albicans* involved in hyphal development, cell wall assembly, and intracellular signaling pathways. Notably, overexpression of HWP1 increased the attachment of *C. albicans* cells to the tooth surface and buccal epithelial cells as the primary stage of colonization and biofilm formation [32,36,37,38].

Results of our study indicated nicotine from all sources led to an increase in the expression level of HWP1 in *C. albicans* planktonic cells with N having the significantly highest effect (Figure 4). The increase in HWP1 expression levels after exposure of *C. albicans* planktonic cells to N has been also described in a previous study [32]. Moreover, an earlier investigation reported a significant increase in the expression level of HWP1, SAP2, and EAP1 genes following exposure of *C. albicans* planktonic cells to CSC, although the study used a predicted ratio of CSC rather than the exact amount of nicotine [22]. In addition, a significant increase in the expression levels of HWP1, SAP2, and EAP1 was identified in the ECSC group compared to the control, ECSC (−) (Figure 4). Studies reported *C. albicans* samples isolated from oral diseases to have significantly higher SAP activity than healthy oral isolates [33,34]. SAP2 is a proteolytic enzyme of *C. albicans* that contributes to adherence, tissue invasion, as well as mucosal and systemic infections [22,37,39]. The participation of HWP1 in *C. albicans* adherence is supported by the EAP1 gene which mediates attachment of *C. albicans* to the surfaces [22,40]. 

Our findings indicated that HWP1 and SAP2 genes overexpressed following exposure to N, CSC, and ECSC (Figure 5). Nevertheless, the expression level of EAP1 and ALS3 between experimental groups was not statistically significant in both *C. albicans* planktonic and biofilm cells (Figure 5). ALS3 is a member of the ALS gene family, which encodes *C. albicans* cell surface adhesion proteins [41]. This protein family mediates attachment and invasion of *C. albicans* to the oral epithelial cells, as well endothelial cells and plays a key role in the biofilm formation of *C. albicans* on prosthetic surfaces [41,42]. Our data indicated that N significantly increased expression levels of the SAP2 gene compared to ECSC in fluconazole treated *C. albicans* planktonic cells. Similarly, the expression levels of the EAP1 and ALS3 genes were elevated in the ECSC group compared to the ECSC (−) control (Figure 5). However, treatment with the antifungal drugs fluconazole or amphotericin B abolished the stimulating effects of CSC/ECSC or ECSC (−) on gene expression in *C. albicans* planktonic cells (Figure 6A,B). Moreover, the presence of fluconazole in sub-MIC concentration seems to interfere with the effect of nicotine from different sources on *C. albicans* biofilm gene expression as none of the targeted genes showed a significant increase following exposure to this antifungal drug. (Figure 7A). Likewise, *C. albicans* biofilm pretreated with amphotericin B did not reveal a significant difference compared to the control group (Figure 7B), however, the expression level of the ALS3 gene was elevated in the ECSC group compared to the ECSC (−) control. Overall, nicotine did not seem to interfere with the effect of antifungal drugs and their impact on virulence features of *C. albicans*. Since this is an in vitro study that did not consider environmental in vivo factors such as dry mouth, etc., the relevance of co-exposure factors needs to be confirmed in clinical studies.

## 5. Conclusions

This study was the first comprehensive investigation to determine the effect of nicotine from different sources on *C. albicans* growth, biofilm formation and in combination with antifungal drug activity. Results of the study indicated that nicotine from different sources affected the pathogenic characteristics of *C. albicans* including hyphal growth, biofilm formation and morphology, and expression of virulence-related genes, which were found differentially expressed. Our findings confirmed that pure N had a different effect on *C. albicans* biofilm formation compared to CSC and ECSC at the same concentration of nicotine.

Our in vitro study indicates that antifungal treatments of *C. albicans* at sub-MIC concentration alleviate the effect of nicotine on virulence-related gene expression. However, the possibility that smoking interferes with different concentrations of antifungal treatments and in vivo studies cannot be completely ruled out. Therefore, further investigation is needed to gain greater insight into the role of cigarettes and e-cigarettes in *C. albicans* pathogenesis and host immune responses.

## Figures and Tables

**Figure 1 ijerph-19-04626-f001:**
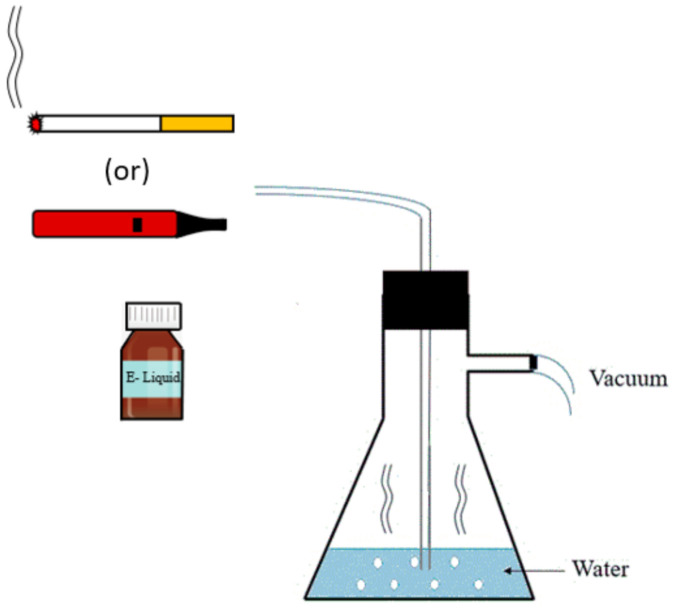
Generation of CSC, ECSC, and ECSC (−). The schematic illustrates the setup that forces smoke from burning cigarettes or e-cigarettes to pass through the water for the preparation of condensates.

**Figure 2 ijerph-19-04626-f002:**
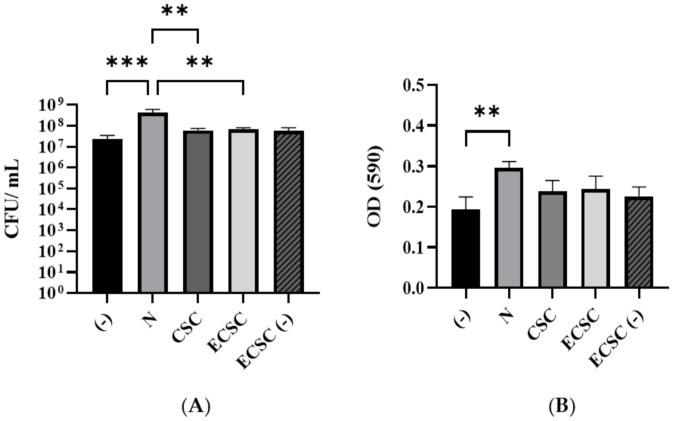
Effect of nicotine from different sources on *C. albicans* biofilm growth was determined by (**A**) CFU and (**B**) MTT assay in YPD medium, ** *p* < 0.01; *** *p* < 0.001. ((−): control, N: pure nicotine, CSC: cigarette smoke condensate, ECSC: e-cigarette smoke condensate with nicotine and ECSC (−): e-cigarette smoke condensate without nicotine).

**Figure 3 ijerph-19-04626-f003:**
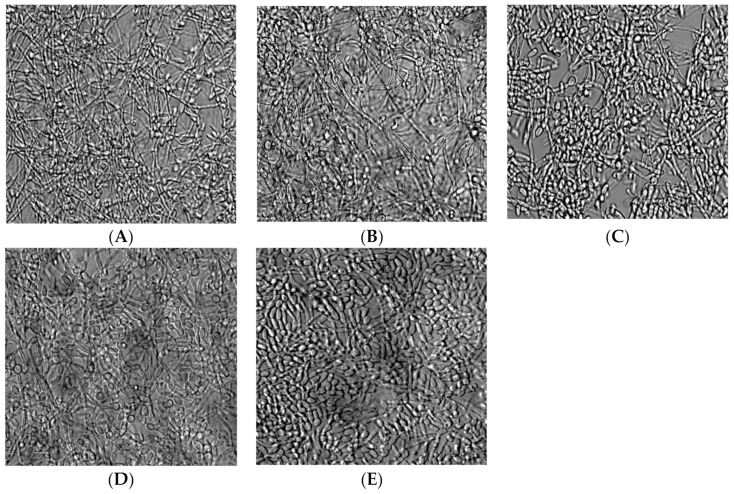
*Candida albicans* biofilm after exposure to nicotine from different sources vs. controls (ECSC (−) and untreated *C. albicans*). N: pure nicotine (**A**), CSC: cigarette smoke condensate (**B**), ECSC: e-cigarette smoke condensate with nicotine (**C**), ECSC (−): e-cigarette smoke condensate without nicotine (**D**) and untreated *C. albicans* (**E**). Images were taken with an inverted microscope through a 20× objective. The scale bar is 100 µm.

**Figure 4 ijerph-19-04626-f004:**
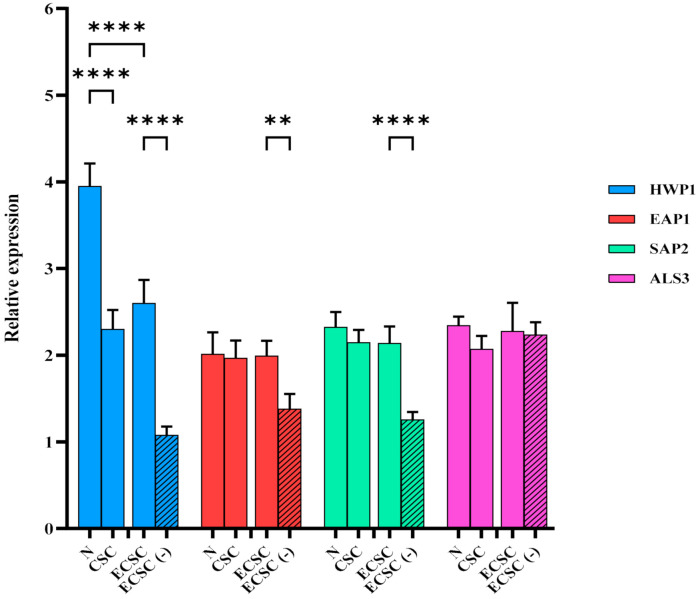
Planktonic *C. albicans* gene expression after the exposure of nicotine from different sources (CSC and ECSC vs. N and ECSC (−)). Expression of the individual genes was normalized to the expression of ACT1, a reference gene, and is shown relative to untreated control cells, ** *p* < 0.01; **** *p* < 0.0001. (N: pure nicotine, CSC: cigarette smoke condensate, ECSC: e-cigarette smoke condensate with nicotine and ECSC (−): e-cigarette smoke condensate without nicotine).

**Figure 5 ijerph-19-04626-f005:**
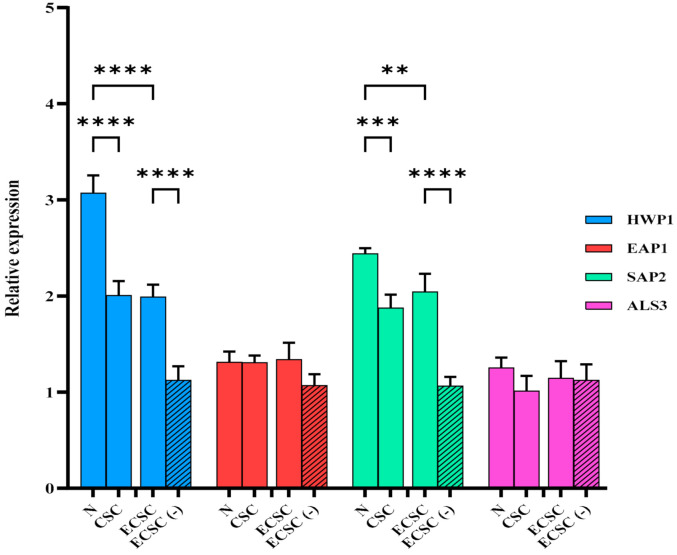
*Candida albicans* biofilm gene expression after exposure to CSC and ECSC vs. N and ECSC (−). Biofilms were formed in a 24-well plate and incubated for 18 h at 37 °C. Expression of the individual genes was normalized to the expression of ACT1, a reference gene, and is shown relative to untreated control cells, ** *p* < 0.01; *** *p* < 0.001; **** *p* < 0.0001. (N: pure nicotine, CSC: cigarette smoke condensate, ECSC: e-cigarette smoke condensate with nicotine and ECSC (−): e-cigarette smoke condensate without nicotine).

**Figure 6 ijerph-19-04626-f006:**
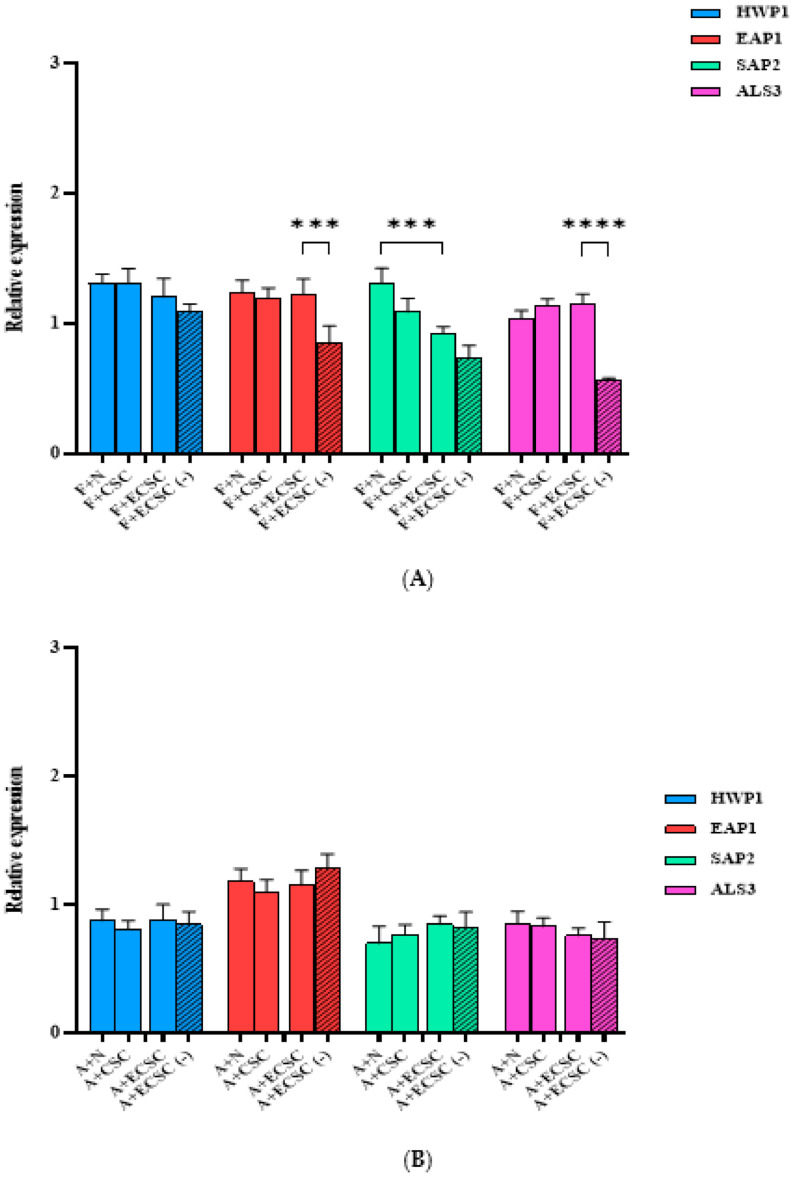
Gene expression of planktonic *C. albicans* after exposure to nicotine only vs. nicotine + antifungal drugs (Flu/AmB)). Planktonic *C. albicans* exposed to 0.1 mg/mL nicotine from different sources and co-incubated with (**A**) sub-MIC concentration of fluconazole, 4 µg/mL and (**B**) 0.1 µg/mL amphotericin B for 18 h at 37 °C. Expression of the individual genes was normalized to the expression of ACT1, a reference gene, and is shown relative to untreated control cells, *** *p* < 0.001; **** *p* < 0.0001. (F: fluconazole, A: amphotericin B, N: pure nicotine, CSC: cigarette smoke condensate, ECSC: e-cigarette smoke condensate with nicotine and ECSC (−): e-cigarette smoke condensate without nicotine).

**Figure 7 ijerph-19-04626-f007:**
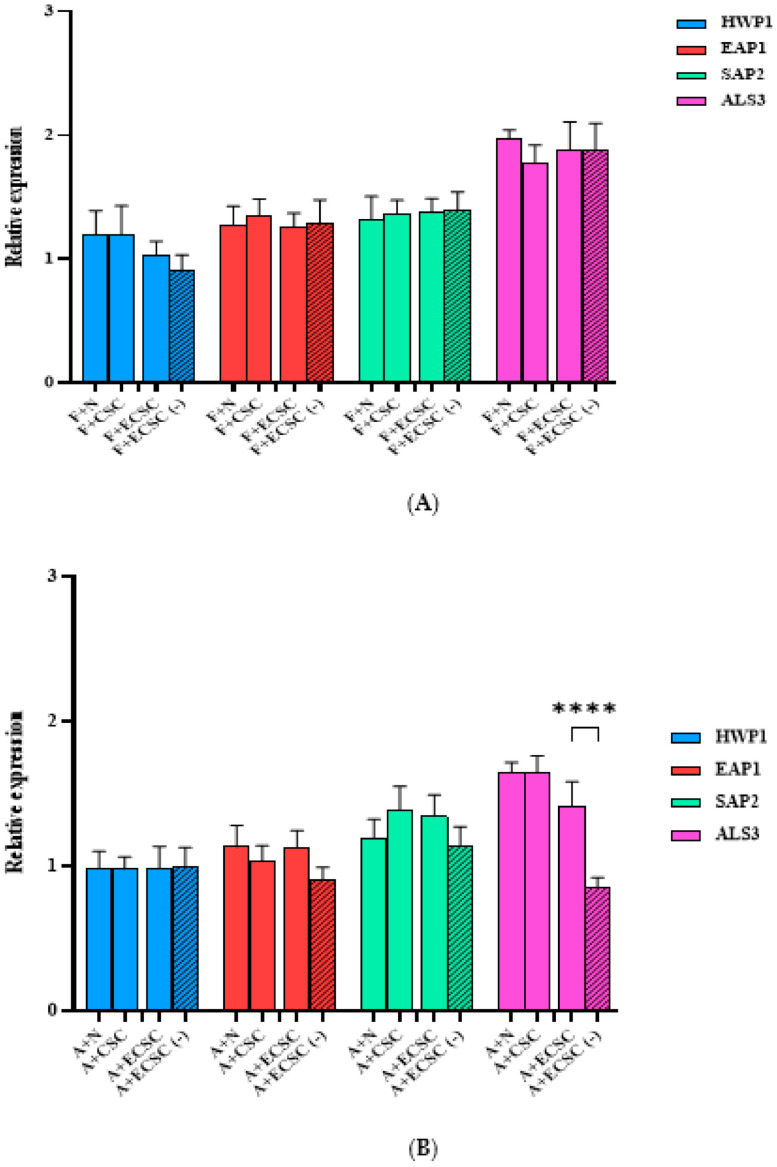
Gene expression of *C. albicans* biofilms after exposure to nicotine only vs. nicotine in the presence of the antifungal drugs fluconazole or amphotericin B. *Candida albicans* biofilm treated with 0.1 mg/mL nicotine from different sources and co-incubated with (**A**) fluconazole (4 µg/mL fluconazole) and (**B**) amphotericin B (0.1 µg/mL). Expression of the individual genes was normalized to the expression of ACT1, a reference gene, and is shown relative to untreated control cells, **** p<0.0001. (F: fluconazole, A: amphotericin B, N: pure nicotine, CSC: cigarette smoke condensate, ECSC: e-cigarette smoke condensate with nicotine and ECSC (−): e-cigarette smoke condensate without nicotine).

**Table 1 ijerph-19-04626-t001:** MIC and MFC of nicotine from different sources (N, CSC, ECSC, and ECSC (−)) and the antifungal drugs (fluconazole and amphotericin B) against *C. albicans*. (N: pure nicotine, CSC: cigarette smoke condensate, ECSC: e-cigarette smoke condensate with nicotine, ECSC (−): e-cigarette smoke condensate without nicotine).

**(A)**		
Groups	MIC (mg/mL)	MFC (mg/mL)
N	4	8
CSC	0.25	0.5
ECSC	0.5	1
ECSC (−)	-	-
**(B)**		
Groups	MIC (µg/mL)	MFC (µg/mL)
Fluconazole	8	16
Amphotericin B	0.25	1

## Data Availability

The data presented in this study are available on request from the corresponding author.

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
