# Peer review of "Effect of Cigarette and E-Cigarette Smoke Condensates on Candida albicans Biofilm Formation and Gene Expression"

_ijerph, 2022, doi:10.3390/ijerph19084626_

Round 1

Reviewer 1 Report

The manuscript should be improved. Here is several major/minor points:

  1. Methodology is not clear yet. For example, no details about the microscopic observation. How the authors analyze the sample (intact or suspension)? If suspended cell used here, please discuss the limitation of this approach.
  2. Following the author’s response, “20 cigarettes are very similar to 20 ml of e-liquid and final nicotine concentration of CSC and ECSC were 3.04 and 2.99 mg/ml respectively.” should be added in the text accordingly.
  3. Yet, the methods about MIC and MFC experiments are not clearly written.
  4. Please make sure the statistical analyses: ANOVA for multiple comparison or a pair comparison? It looks the t-test. In addition, in Figure 2, are there any significant differences among (-) control, CSC, ECSC, ECSC(-) (Panels A&B)? Same issue in all the data presented as bar graphs.
  5. qPCR data presentation; is there any reasons for presenting the data as delta/delta CT? How about the expression of genes relative to control group?
  6. I have no idea on the overexpression of HWP1, SAP1. Compared with which group? Please clearly write which group is used for the comparison.
  7. Antifungal drug treatment: Since fluconazole is a fungistatic drug, did authors check the cell growth/morphology after treatment with/without cigarette/E-cigarette smoke condensates? If the cell is not active, obviously the gene expression is not effected by nicotine. Please provide detailed data here.

Author Response

1. Methodology is not clear yet. For example, no details about the microscopic observation. How the authors analyze the sample (intact or suspension)? If suspended cell used here, please discuss the limitation of this approach.

Details of the microscopic observations have been added.

2. Following the author’s response, “20 cigarettes are very similar to 20 ml of e-liquid and final nicotine concentration of CSC and ECSC were 3.04 and 2.99 mg/ml respectively.” should be added in the text accordingly.

This detail has been added to the text as suggested.

3. Yet, the methods about MIC and MFC experiments are not clearly written.

The MIC and MFC have been modified to better reflect the methods used in the study.

4. Please make sure the statistical analyses: ANOVA for multiple comparison or a pair comparison? It looks the t-test. In addition, in Figure 2, are there any significant differences among (-) control, CSC, ECSC, ECSC(-) (Panels A&B)? Same issue in all the data presented as bar graphs.

Statistical analysis of differences between the control (non-treated) and the test (treated) samples was performed using the two-way ANOVA function in GraphPad. Multiple comparisons have been added to the material and methods.

All data represent the relative expression of the respective genes in the test groups compared to control (-) group.

5. qPCR data presentation; is there any reasons for presenting the data as delta/delta CT? How about the expression of genes relative to control group?

The relative gene expression compared to the control (-) group has been added to qPCR graphs.

6. I have no idea on the overexpression of HWP1, SAP1. Compared with which group? Please clearly write which group is used for the comparison.

Data shows the relative expression of the genes in different groups compared with the control (-). A clarification has been added to the figure legends.

7. Antifungal drug treatment: Since fluconazole is a fungistatic drug, did authors check the cell growth/morphology after treatment with/without cigarette/E-cigarette smoke condensates? If the cell is not active, obviously the gene expression is not effected by nicotine. Please provide detailed data here.

Cell viability data have been added as supplemental data.

Reviewer 2 Report

The authors have responded to the comments.  Amendment has been made according to the suggestion.  No further comment from the reviewer. 

Author Response

Thank you so much

This manuscript is a resubmission of an earlier submission. The following is a list of the peer review reports and author responses from that submission.

Round 1

Reviewer 1 Report

The manuscript presents an interesting and important health issue related influence of electronic cigarettes on oral health. Such research is essential to understand the impact of e-cigarettes on the oral cavity, which, in the light of previous research arose more and more controversy. This manuscript is well written and organized. However, I would like to suggest the following points for the betterment of this manuscript:

1.In the Introduction section I suggest to include recently publisher paper: CichoÅ„ska D, Kusiak A, Piechowicz L, Åšwietlik D. A pilot investigation into the influence of electronic cigarettes on oral bacteria. Advances in Dermatology and Allergology 2021; 38(6):1092-1098. doi:10.5114/ada.2020.100335. which also concerns the influence of e-cigarette usage on C.albicans in the oral cavity.

2.At the beginning of the Materials and methods section I suggest to to explain the abbreviation CSC, ECSC and ECSC (-) to make content of the publication  more understandable for recipients who are less familiar with the subject of the conducted research. 

3.Figures clearly present the results of the conducted research, however tables should be graphically adapted to the journal's guidelines. 

4.References should be provided with all authors names, not only the first author. 

Author Response

1. In the Introduction section I suggest to include recently publisher paper: CichoÅ„ska D, Kusiak A, Piechowicz L, Åšwietlik D. A pilot investigation into the influence of electronic cigarettes on oral bacteria. Advances in Dermatology and Allergology 2021; 38(6):1092-1098. doi:10.5114/ada.2020.100335. which also concerns the influence of e-cigarette usage on C.albicans in the oral cavity. 

The article has been included in the introduction as suggested.

2. At the beginning of the Materials and methods section I suggest to explain the abbreviation CSC, ECSC and ECSC (-) to make content of the publication  more understandable for recipients who are less familiar with the subject of the conducted research. 

Explanations of the abbreviations have been added.

3.Figures clearly present the results of the conducted research, however tables should be graphically adapted to the journal's guidelines.

Tables have been corrected according to the journal guidelines.

4. References should be provided with all authors names, not only the first author. 

References have been corrected as suggested.

Reviewer 2 Report

Reviewer 

This is a very interesting paper, but in vivo research is also desired. .  Please find below the suggestions for the manuscrript:1. Did you consider evaluation such as C.glabrata? 

2.  It is also necessary to describe environmental factors as a study limitation. Dry mouth etc. 3. Conclusion can be better written.

Author Response

This is a very interesting paper, but in vivo research is also desired.  Please find below the suggestions for the manuscript:

This is a great suggestion, but beyond the scope of the current study. We will consider expanding to in vivo work in future studies.

1. Did you consider evaluation such as C.glabrata?

This is an interesting and relevant suggestion, which we will consider for future study as inclusion in the current manuscript is beyond its scope.

  1.  It is also necessary to describe environmental factors as a study limitation. Dry mouth etc.

Environmental factor have been added to the discussion of study limitations.

3. Conclusion can be better written.

The conclusion has been modified to better reflect the findings of the study.

Reviewer 3 Report

The manuscript describes the effect of cigarette and e-cigarette smoke condensates on Candida albicans biofilm formation and gene expression.  The methodology is robust.

Specific comments

Page 1 line 19 – ‘biofilm’

Page 2 line 59 – CSC and ECSC should firstly be written in full in the introduction even it has been explained in the abstract.

Page 2 line 65 – How did you prepare ECSC (-)?  IR3F is CSC, 0 mg is ECSC and 24 mg nicotine is ECSC (-)? Seems to contradict the abstract.  Please clarify.

Page 2 line 68 – ‘standard’

Page 2 line 70 - respective

Page 2 line 73 – ‘Technologies’

Page 3 line 90, line 93 and line 103 – Candida albicans should be written as C. albicans

Page 3 line 101 – ‘hr’ should be written a ‘h’.

Page 3 line 99 to 111 – Please justify why do you choose 0.1 mg/mL?  Also, please justify why the biofilm was developed only for 18 h?  The maturation of biofilm is normally at 72 h.

Page 4 line 112 to 125 – Please explain how do you culture planktonic C. albicans.

Page 4 line 115 – Is it a sterile PBS?

Page 4 line 129 to 139 – Please standardise either to write the numbers (1 to 10) alphabetically or numerical.

Page 4 line 144 – How many biological and technical replicates?

Page 5 line 173 – Do you have results for ‘Candida albicans without cigarette and e-cigarette’ as a positive control?

Page 5 line 177 – What is the P-value when you mention the significantly enhanced biofilm formation?

Page 6 line 191 – What is the P-value for the significantly increased HWP1 gene expression?

Page 7 line 216 – What is the P-value for significantly different expressions?

Page 10 line 278 – What do you mean by minor effect?

Page 10 line 288 – I am not sure how do you conclude that N significantly increases the biofilm of C. albicans when you don’t have untreated C. albicans? What is the comparison that you are referring here? Please clarify.

Author Response

Page 1 line 19 – ‘biofilm’

This has been corrected.

Page 2 line 59 – CSC and ECSC should firstly be written in full in the introduction even it has been explained in the abstract.

Full names of CSC and ECSC have been added to the introduction.

Page 2 line 65 – How did you prepare ECSC (-)?  IR3F is CSC, 0 mg is ECSC and 24 mg nicotine is ECSC (-)? Seems to contradict the abstract.  Please clarify.

Standard IR3F cigarettes were used to prepare CSC.  Two different e-cigarette liquids containing 0 and 24 mg nicotine were used to prepare e-cigarette smoke condensates with nicotine (ECSC) and without nicotine,( ECSC (-)).

Page 2 line 68 – ‘standard’

This has been corrected.

Page 2 line 70 – respective

This has been corrected.

Page 2 line 73 – ‘Technologies’

This has been corrected.

Page 3 line 90, line 93 and line 103 – Candida albicans should be written as C. albicans

Since the genus name is at the beginning of the sentence it has been spelled out.

Page 3 line 101 – ‘hr’ should be written a ‘h’.

 This has been corrected.

Page 3 line 99 to 111 – Please justify why do you choose 0.1 mg/mL?  Also, please justify why the biofilm was developed only for 18 h?  The maturation of biofilm is normally at 72 h.  

0.1 mg/mL concentration is the sub-MIC nicotine concentration which still allows normal C. albican growth for nicotine derived from the different sources evaluated in this study.

In this study we have tested early-stage of C. albicans biofilm and will consider evaluating the effect of nicotine from different sources on mature biofilms for future study.

Page 4 line 112 to 125 – Please explain how do you culture planktonic C. albicans.

The information has been added to the mansucript. For liquid planktonic culture growth, one colony of C. albicans was inoculated into 10 mL YPD broth.

Page 4 line 115 – Is it a sterile PBS?

 Yes, the PBS used is sterile. This has been added.

Page 4 line 129 to 139 – Please standardize either to write the numbers (1 to 10) alphabetically or numerically.

This has been corrected. Numbers 1 to 10 are written alphabetically.

Page 4 line 144 – How many biological and technical replicates?

3 biological and 3 technical replicates,(three separate experiments in triplicate).

Page 5 line 173 – Do you have results for ‘Candida albicans without cigarette and e-cigarette’ as a positive control?

Yes, positive control is shown as (-) in the figures.

Page 5 line 177 – What is the P-value when you mention the significantly enhanced biofilm formation?

The p-value of < 0.0001 for significantly enhanced biofilm formation has been added.

Page 6 line 191 – What is the P-value for the significantly increased HWP1 gene expression?

The corresponding p-value of < 0.0001 has been added.

Page 7 line 216 – What is the P-value for significantly different expressions?

The corresponding p-value of < 0.0001 has been added.

Page 10 line 278 – What do you mean by minor effect?

 This has been reworded and now reads “resulted in a small non-significant reduction" to clarify the findings of the cited study.

Page 10 line 288 – I am not sure how do you conclude that N significantly increases the biofilm of C. albicans when you don’t have untreated C. albicans? What is the comparison that you are referring here? Please clarify.

Untreated C. albicans (control) is shown as (-) in all figures.

Reviewer 4 Report

The authors investigated effect of cigarette and E-cigarette smoke condensates on C. albicans biofilm formation. The authors also trying to evaluate antifungal drug interaction (not comprehensively) with nicotine (compared to nicotine-free) and the effect on C. albicans biofilm formation and genes expression.

The major concerns of this study:

  1. Rationale of this study; need to clarify why nicotine and antifungal drug are used in this study. It is just mentioned in the conclusion part.
  2. Please provide the HPLC chromatograms (standard, cigarette, E-cigarette, nicotine-free E-cigarette) indicating the target peak. This would be also helpful to characterize the differences between cigarette and E-cigarette (or E-cigarette vs. nicotine-free E-cigarette).
  3. Please describe the details on the preparation of cigarette condensates (here the authors use the vacuum in the totally different type of cigarettes). Also wondering whether 20 cigarettes are equivalent to 20 ml of e-liquid.
  4. No details on MIC and MFC experiments
  5. Most critical point of this study is that the authors need to present the morphologic differences of C. albicans in the presence of smoke condensates. This is essential to understand the gene expression corresponding to hyphal formation. Thus, the reviewer strongly suggests the biofilm imaging.
  6. It would be interesting to consider whether smoke condensates affect the gingival epithelial cells/or interacting with C. albicans.
  7. Data presentation; Figure title in the bar graphs (gene expression; nicotine only vs. nicotine + antifungal drugs (Flu/AmB)) would be helpful.
  8. Current experimental design could not be reached to the antifungal drug-nicotine interactions indicating how antifungal drug (sub-MIC; one conc. combination) alleviate the effect of nicotine.
  9. Conclusion is not clearly written.

Minor:

  1. Line 2: italicize scientific names (Candida albicans) here and throughout the manuscript
  2. Please check typos; for example, CO2 and hyphen use

Author Response

Rationale of this study; need to clarify why nicotine and antifungal drug are used in this study. It is just mentioned in the conclusion part.

The following sentence has been added to the last paragraph of the introduction to highlight the importance of investigating the effect of nicotine on antifungal drugs efficacy: “….. despite the heightened risk of smokers for oral candidiasis [3,4], it is currently unclear if nicotine affects the efficacy of antifungal treatments.”

Please provide the HPLC chromatograms (standard, cigarette, E-cigarette, nicotine-free E-cigarette) indicating the target peak. This would be also helpful to characterize the differences between cigarettes and E-cigarette (or E-cigarette vs. nicotine-free E-cigarette).

The HPLC chromatograms have been added as supplemental material (Figure A1).

Please describe the details on the preparation of cigarette condensates (here the authors use the vacuum in the totally different type of cigarettes). Also wondering whether 20 cigarettes are equivalent to 20 ml of e-liquid. Process of extracting cigarette and e-cigarette were exactly the same. 

Smoke was extracted by inserting cigarette/e-cigarette into the end of the plastic tube connected to a vacuum flask and pulling down the cigarette/e-cigarette smoke into the water as depicted in Figure 1. Also, 20 cigarettes are very similar to 20 ml of e-liquid and final nicotine concentration of CSC and ECSC were 3.04 and 2.99 mg/ml respectively.

No details on MIC and MFC experiments. 

The experimental details have been added.

Most critical point of this study is that the authors need to present the morphologic differences of C. albicans in the presence of smoke condensates. This is essential to understand the gene expression corresponding to hyphal formation. Thus, the reviewer strongly suggests the biofilm imaging.

This is an interesting suggestion and images with representative biofilm cell morphologies in the presence and absence of nicotine from the different sources investigated in this study have been added to the manuscript as figure 3.

It would be interesting to consider whether smoke condensates affect the gingival epithelial cells/or interacting with C. albicans.

This has been investigated and will be presented in our future manuscript that is currently in preparation.

Data presentation; Figure title in the bar graphs (gene expression; nicotine only vs. nicotine + antifungal drugs (Flu/AmB)) would be helpful.

This has been corrected as suggested.

Current experimental design could not be reached to the antifungal drug-nicotine interactions indicating how antifungal drug (sub-MIC; one conc. combination) alleviate the effect of nicotine.

We agree with the reviewer that the current experimental design does not elucidate how drug-nicotine interaction alleviate the gene expression modulating effect of nicotine. The corresponding findings in this study are observational but show that antifungal drug treatment at clinically relevant doses can inhibit the increase in expression of the virulence related genes examined in this study.

Conclusion is not clearly written.

The conclusion has been modified to better reflect the findings of the study.

Minor:

Line 2: italicize scientific names (Candida albicans) here and throughout the manuscript:

This has been corrected.

Please check typos; for example, CO2 and hyphen use.

This has been checked and corrected as necessary.